# The Validity of the Original and the Saudi-Modified Screening Tools for the Assessment of Malnutrition in Pediatrics: A Cross-Sectional Study

**DOI:** 10.3390/diagnostics14202256

**Published:** 2024-10-10

**Authors:** Sheikha Alqahtani, Khalid Aldubayan, Saleh Alshehri, Ghada Almuhareb, Amal Mahnashi

**Affiliations:** 1Department of Community Health Sciences, Collage of Applied Medical Sciences, King Saud University, Riyadh 11451, Saudi Arabia444204048@student.ksu.edu.sa (A.M.); 2Department of Dietetics, Prince Sultan Military Medical City, Riyadh 11159, Saudi Arabia; 3Department of Emergency, Prince Sultan Military Medical City, Riyadh 11159, Saudi Arabia; drsaleh2000@yahoo.com; 4Department of Clinical Nutrition, Riyadh Third Health Cluster, Riyadh 13717, Saudi Arabia

**Keywords:** pediatric malnutrition, validation, screening tools

## Abstract

**Background**: Screening for malnutrition among hospitalized children is essential, and the Screening Tool for the Assessment of Malnutrition in Pediatrics (STAMP) is a validated tool for this purpose. The study aimed to modify STAMP for a Saudi context and assess the sensitivity and specificity of both the original and modified tools. **Method**: A cross-sectional study was conducted among 307 hospitalized children, where both the original and Saudi-modified STAMP were applied. Anthropometric measurements were also recorded, and statistical analysis using SPSS and validity parameters was used to assess the tools’ validity. **Results**: The Saudi-modified STAMP identified a higher percentage of children at high risk of malnutrition compared to the original STAMP (91.6% vs. 62.9%). The sensitivity, specificity, positive predictive value, negative predictive value, accuracy, and agreement of the Saudi-modified STAMP compared to the original were 94.3%, 13.2%, 64.8%, 57.7%, 0.654, and 0.089, respectively. **Conclusions**: The Saudi-modified STAMP showed excellent sensitivity and varied negative predictive value, indicating its potential effectiveness in screening for the risk of malnutrition among hospitalized children compared to the original STAMP.

## 1. Introduction

Pediatric malnutrition is a prevalent condition, associated with impaired growth and development, unfavorable clinical outcomes, longer hospital stays, delayed recovery, and increased healthcare costs [1,2]. Consequently, it is imperative to comprehensively assess the nutritional status of pediatric patients, employ various validated methods, and collaborate with a multidisciplinary team of healthcare professionals to deliver optimal nutritional interventions as an integral part of their hospital care [1,2]. Multiple studies have globally reported a wide range in the prevalence of malnutrition among hospitalized children, varying from 6.1% to 55.6% [3,4]. The observed differences in prevalence rates can be attributed to variations in defining criteria and the geographical areas examined. The lack of a standardized definition is responsible for the underestimation of the prevalence of malnutrition; while the World Health Organization (WHO) defined pediatric malnutrition as a nutritional imbalance involving either undernutrition (stunting, wasting, or underweight) or overnutrition (overweight) [5], Mehta and his colleagues define pediatric malnutrition (undernutrition) as an imbalance between nutritional intake and needs, leading to cumulative shortfalls in energy, protein, or micronutrients that may have a deleterious impact on development, growth, and other clinical outcomes [6].

According to the WHO, malnutrition is a significant contributor to childhood mortality, accounting for approximately 45% of all cases [5]. Most medical conditions cause malnutrition through a loss of appetite, disturbing of the digestive system, and/or by increasing nutritional needs [7]. Worldwide, the main causes of mortality come from preventable and/or treatable medical conditions such as preterm birth complications, birth asphyxia/trauma, pneumonia, diarrhea, malaria, and infectious diseases [8]. In Saudi Arabia, respiratory infections, neonatal disorders, pneumonia, and acute diarrhea are recognized as significant contributors to mortality rates, potentially influencing the nutritional status of individuals [9,10].

Evaluation of the nutritional status of hospitalized children is strongly recommended by internationally recognized organizations such as the European Society for Pediatric Gastroenterology, Hepatology, and Nutrition, the European Society for Clinical Nutrition and Metabolism (ESPEN), and the American Society for Parenteral and Enteral Nutrition (ASPEN) [11,12,13]. Indeed, numerous tools have been developed and validated to screen pediatric risk of malnutrition in hospital settings. These tools aim to assist healthcare professionals in screening for malnutrition risk among hospitalized children [14,15,16]. The Screening Tool for the Assessment of Malnutrition in Pediatrics (STAMP) was specifically designed to be incorporated into the admission screening procedure for pediatric patients. One of the strengths of STAMP is that it can be completed by any healthcare professional, even without formal nutrition training. This feature allows for its widespread use within hospital settings, ensuring that malnutrition screening becomes a routine part of patient care [17,18]. STAMP was developed and validated by a team from the Royal Manchester Children’s Hospitals and the University of Ulster. It is a simple five-step tool designed for screening malnutrition on admission and during the hospitalization of children aged 2 to 16 years [17,19]. This tool was created to provide efficient and effective means of identifying malnutrition risk in pediatric patients [17,19].

In addition, the validation of screening tools commonly involves the use of reference standards, such as anthropometric measurements, dietary intake assessments, and nutrition-related biochemical markers [20]. These indicators are considered to be the most common standards for detecting pediatric patients at risk of malnutrition [21]. The Academy of Nutrition and Dietetics and ASPENs recommended the following anthropometric measurements (weight-for-height Z-score, body mass index for age Z-score, length/height for age Z-score, and mid-upper arm circumference) as primary indicators to diagnose pediatric malnutrition [22].

Based on the literature review conducted, and to the best of our knowledge, there are limited studies that have modified and evaluated the sensitivity and specificity of modified nutrition screening tools using anthropometry measurements as reference standards [23]. Additionally, no previous studies have been conducted in Middle Eastern, specifically Saudi Arabian, contexts to modify and evaluate the sensitivity and specificity of nutrition screening tools in detecting malnutrition.

A Saudi-modified STAMP was developed by healthcare providers at the Prince Sultan Military Medical City, incorporated with King Saud University, by modified diagnosis, in part by adding the main medical conditions that contributed to childhood mortality among Saudi children; as a definite nutritional implication, respiratory infection and neonatal disorders were added; and as a possible nutritional implication, pneumonia and acute diarrhea [9,10] were added. The modification in weight and height for age was calculated by using the centile quick-reference tables based on SGCs [24]. Therefore, this study aims to assess the validity of the original STAMP, and the Saudi-modified STAMP, in detecting the risk of malnutrition upon admission among Saudi pediatric patients. The original STAMP, along with anthropometric measurements, and dietary intake assessments using 24 h dietary recall method, will be used as reference standards in this evaluation.

## 2. Materials and Methods

### 2.1. Study Design and Setting

Cross-sectional evaluations of studies of diagnostic tools assessed the validity of the Saudi-modified Screening Tool for the Assessment of Malnutrition in Pediatrics (S-mSTAMP) and the original Screening Tool for the Assessment of Malnutrition in Pediatrics (oSTAMP) on admission using anthropometric measurements as reference standards to assess the malnutrition risk among Saudi hospitalized children at Prince Sultan Military Medical City (PSMMC).

### 2.2. Study Population

The sample size was calculated based on the sensitivity (Se), specificity (Sp), and prevalence of malnutrition as found in Perez Solis et al.’s paper [25], by using the following formulas: (1) *n* = (z^2^ × p(1 − p))/∆^2^; *n* will be (a + c) if we used Se as p, and *n* will be (b + d) if we use Sp as p, (2) *n* = ((a − c))/prevalence, (3) *n* = ((b + d))/((1 − prevalence)). The total sample size required for this study was 292. The inclusion criteria were as follows: Saudi children of both genders aged from six months to 14 years who are admitted to the emergency ward at PSMMC. The exclusion criteria were children with edema, ascites, fluid retention, major congenital anomalies (heart defects, neural tube defects, and Down syndrome), and non-Saudi children.

### 2.3. Ethical Board Approval

This study was conducted at PSMMC, Riyadh, Saudi Arabia between November 2022 and November 2023. The study was approved by the Research Ethics Committees at PSMMC (Institutional Review Board Approval No: E2001; Date of approval: 8 November 2022) and at King Saud University (Institutional Review Board Approval No: E-22-7184; Date of approval: 24 November 2022).

### 2.4. Sociodemographic Data and Medical Variables of the Patient

Sociodemographic variables regarding the parents were taken by the researchers through interviews, including sex (Male or Female), age, education (None, Elementary, Secondary, High school, Graduate, Postgraduate), city of origin of parents, marital status (Married, Divorced, Widowed), number of children, occupation (Homemaker, Office employee, Trader, Worker, Military), and monthly household income by Saudi riyal (SR) (Less than 5000 SR, 5000–10,000 SR, 11,000–20,000 SR, More than 20,000 SR); also, sociodemographic variables regarding the children were taken by the researcher through interviews, including age and sex (Boy, Girl) [26]. Medical variables of the patient were taken from the medical file, including the diagnosis, amount of time hospitalized, and time since diagnosis [26].

### 2.5. Anthropometric Measurements

Clinical nurses and researchers performed the measurements using standardized methods as per hospital policy. Weight, length or height, and mid-upper arm circumference (MUAC) were measured, and body mass index (BMI) and growth parameters were statistically measured and presented as Z-scores. Weight: all subjects aged less than one year were weighed on DETECTO’s MB130 digital scale (Webb City, MO, USA), and those aged more than one year were weighed using a Digital Pearson Scale with a precision of 10 g on admission. The length or height was taken on admission using measurement tape for patients aged less than two years, and for those aged over two years, the Digital Pearson Scale was used, with patients standing and facing the scale, to an approximate 0.5 cm without shoes. BMI was calculated by dividing the weight in kilograms by the square of the height in meters (kg/m²). Growth parameters included weight-for-length/height Z-scores for children aged less than five years, and BMI-for-age Z-scores for children older than five years old were calculated using World Health Organization growth charts (WHOGCs) for children younger than 2 years, Central for Disease Prevention and Control growth charts (CDCGCs) for children older than 2 years, and Saudi growth charts (SGCs) for children aged from 6 months to 14 years. Then, the malnutrition cutoffs were determined for each patient based on the Academy of Nutrition and Dietetics/American Society of Parenteral and Enteral Nutrition (ASPEN) 2014 Pediatric Malnutrition Consensus Statement. Based on weight-for-length/height Z-scores and BMI-for-age Z-scores, patients were classified as overweight/obese/very obese/normal weight with Z-scores between ≥3 and −0.99, and as having mild malnutrition from −1 to −1.99, moderate malnutrition from −2 to −2.99, and severe malnutrition at ≥−3. Patient nutrition status was consolidated into two groups: “malnutrition” (mild malnutrition, moderate malnutrition, and severe malnutrition), and “absence of malnutrition” (overweight/obese/very obese/normal) [22,27].

MUAC was measured by researchers within 48 h of admission. First, the midpoint of the left upper arm was determined (between the tip of the shoulder and the tip of the elbow), then tape was used to determine the MUAC value; and finally, we classified the nutritional status depending on MUAC cutoffs, which differed according to age groups: children aged between 6 and 59 months old with MUAC < 115 mm, 5–9 years < 135 mm, or 10–14 years < 160 mm were considered to suffer from severe acute malnutrition; and children aged between 6 and 59 months ≥ 115 and <125 mm, or 5–9 years ≥ 135 and <145 mm, 10–14 years ≥ 160 and <185 mm were considered to suffer from moderate acute malnutrition [28].

### 2.6. Nutritional Screening

#### 2.6.1. The Original Screening Tool for the Assessment of Malnutrition in Pediatrics (oSTAMP)

As a routine policy, the original STAMP was applied by clinical nurses who completed a nutrition screening at the time of admission by assessing nutrition risk based on 3 steps: each step was scored out of three (see Appendix A). First, it scores the nutritional implications of the admission diagnosis, where 3 indicated definite nutritional implications, 2 was possible nutritional implications, and 0 was no nutritional implications. Then, it assessed the nutritional intake, where no change in eating patterns and good nutritional intake scored 0, recently decreased or poor nutritional intake scored 2, and no nutritional intake scored 3. Lastly, weight and height-for-age were assessed using the centile quick-reference tables, where 0 to 1 centile spaces/columns apart scored 0, >2 centile spaces/=2 columns apart scored 1, and >3 centile spaces/≥3 columns apart (or weight < 2nd centile) scored 3. The overall risk of malnutrition was categorized into three categories based on a total score from the previous three steps: at high risk if equal to or more than 4, at medium risk if the score was between 2 and 3, or at low risk if the score was between 0 and 1 [17,19].

#### 2.6.2. Saudi Modified Screening Tool for the Assessment of Malnutrition in Pediatrics (S-mSTAMP)

After the clinical nurses applied the original STAMP, the researchers (Dietitians) applied the Saudi-modified STAMP. The modification in the original STAMP was in two out of three steps (see Appendix A). First, in the diagnosis step, the main medical conditions that contributed to childhood mortality among Saudi children as definite nutritional implications were added, and respiratory infection and neonatal disorders, as well as pneumonia and acute diarrhea as possible nutritional implications [9,10]. The modification in weight and height for age was calculated using the centile quick-reference tables based on SGCs [24].

### 2.7. Statistical Analysis

The statistical analysis was performed using IBM SPSS Statistics for Windows (version 26; IBM Corp., Armonk, NY, USA). The normality of all the quantitative variables was tested before performing the analysis using the Shapiro–Wilk test. Missing data were treated using Mean/Median/Mode imputation. Descriptive analysis results for continuous data were shown as means ± standard divisions (SD) for normally distributed data or median and interquartile range (IQR) for data not normally distributed. Descriptive analysis results for categorical data were shown as frequencies and percentages. The chi-square or Fisher test was used for categorical variables. For continuous data, Student’s *t*-test was used for normally distributed variables, and the Mann–Whitney U test was used for non-normally distributed variables to compare patients at low risk of malnutrition with patients at high risk of malnutrition. A *p*-value of <0.05 was used to report the statistical significance and precision of the estimates.

The screening tool was validated using the area under the ROC curve (AUC), sensitivity (Se), specificity (Sp), negative (NPV), and positive (PPV) predictive value. Se and NPV were given more weight when validating screening tools. Se and Sp values were rated as follows: >90 excellent, 80 to 90 good, 70 to 80 fair, 60 to 70 insufficient, and 50 to 60 poor, for which the overall degrees of Se, SP, PPV, and NPV were high > 90, moderate 80 to 90, and low < 80 [29]. The agreement between the Saudi-modified STAMP and the original STAMP was analyzed by the kappa (κ) value. κ values were rated as follows: >0.90 almost perfect, 0.8 to 0.9 strong, 0.6 to 0.79 moderate, 0.40 to 0.59 weak, 0.21 to 0.39 minimal, 0 to 0.20 none, for which the overall level of agreement was high if >0.8, moderate at 0.6 to 0.79, or low at <0.59 [30].

## 3. Results

### 3.1. Sociodemographic Characteristics

Sociodemographic data of children’s caregivers were taken, 256 females, 83.4%, and 51 males, 16.6%, were included; the mean age was 36.4 years old. The majority were university graduates; almost all of them were living in Riyadh, married, and homemakers. Monthly family income from SAR 5000 to 10,000 was common, representing 45.6 % of our study population.

A total of 307 hospitalized children were included in the analysis: 139 girls, 45.3%, and 168 boys, 54.7%; with the age ranging from 6 to 170 months (median 59 months, interquartile range (IQR) 72 months). Under investigation and asthma were considered the main admission diagnoses documented in the patient’s medical records, presented as 57.7% and 33.9%, respectively. The main medical condition causing childhood mortality in Saudi Arabia, pneumonia, was considered the main admission diagnosis documented in our study, at 12.4%; this was followed by respiratory infection, acute diarrhea, and neonatal disorder, as shown in (Table 1).

### 3.2. Nutrition Status Characteristics Based on Anthropometric Measurements

The nutritional status based on anthropometric measurements is presented in Table 2. Most patients presented with a normal nutritional status based on World Health Organization growth charts (WHOGCs), Central for Disease Prevention and Control growth charts (CDCGCs), and Saudi growth charts (SGCs) for both weight-for-height Z-scores and Body Mass Index (BMI)-for-age Z-scores, but with Saudi BMI-for-age Z-scores most of the patients were overweight (see Appendix A). We categorized the patient’s nutritional statuses into two groups where we combined all undernutrition categories under the malnourished group, named malnutrition, and combined the normal categories with all overnutrition categories, named absence of malnutrition.

### 3.3. Nutritional Status Characteristics Based on the Original and the Saudi Modified Screening Tool for the Assessment of Malnutrition in Pediatrics (STAMP)

Table 3 presents a summary of each part of STAMP, which shows most patients’ diagnoses were matched with no nutritional implications based on both the Original Screening Tool for the Assessment of Malnutrition in Pediatrics (oSTAMP) and The Saudi Modified Screening Tool for the Assessment of Malnutrition in Pediatrics (S-mSTAMP). Most of our sample had no change in eating patterns and good nutritional intake based on oSTAMP, but recently decreased or poor nutritional intake based on S-mSTAMP. Additionally, most of the patients based on oSTAMP were at 0 to 1 centile spaces/columns apart, while, based on S-mSTAMP, they were at >3 centile spaces/≥3 columns apart (or weight < 2nd centile). Finally, most of the patients were classified as having medium risk of malnutrition based on the oSTAMP, while being at high risk of malnutrition based on S-mSTAMP. The nutrition risks were presented as “high risk” or “low risk”; in the “high-risk group” high and medium risk were combined, while “low risk” was presented alone. The rationale for combining the “high” and “medium” risk data into a single “high-risk group” in the analysis was based on literature that supports this categorization. We believe that this approach facilitates a clearer understanding of the risk levels and allows for more effective action in comparison to the “low risk” group. 

### 3.4. Anthropometric Characteristics of Hospitalized Children According to Original and Saudi Modified Screening Tool for the Assessment of Malnutrition in Pediatricsc (STAMP)

Regarding the anthropometric measurements, there was no significant difference between the two groups (at high risk and at low risk) when using the oSTAMP or S-mSTAMP (Table 4).

Growth status Z-score parameters defined nutritional status; most patients presented with an absence of malnutrition based on WHOGCs, CDCGCs, and SGCs, compared with malnutrition as presented based on weight-for-height Z-scores using WHOGCs, CDCGCs, and SGCs; based on BMI-for-age Z-scores using CDCGCs, and SGCs; and based on mid-upper arm circumference (MUAC).

Fisher’s exact test suggests that there is no significant difference between the nutritional status based on growth status Z-score parameters, including Saudi weight-for-height, Saudi BMI-for-age, WHO and CDC weight-for-height, and CDC BMI-for-age Z-scores, and the nutritional status based on S-mSTAMP; but, the number of patients at high risk of malnutrition according to S-mSTAMP was higher than patients at low risk based on Saudi weight-for-height, Saudi BMI-for-age, WHO and CDC weight-for-height, and CDC BMI-for-age Z-scores, compared with patients at low risk. The average MUAC was 153.37 ± 40.69. Also, the *t*-test suggests that there are no significant differences between patients presenting with malnutrition and absence of malnutrition both based on oSTAMP and S-mSTAMP, with *p* = 0.961 and *p* = 0.621, respectively. 

### 3.5. Prevalence of Patients at High Risk of Malnutrition, Validity, and Agreements of Saudi Modified Screening Tool for the Assessment of Malnutrition in Pediatrics Using Original Screening Tool for the Assessment of Malnutrition in Pediatrics

The prevalence of patients at high risk of malnutrition based on S-mSTAMP diagnosis, nutritional intake, weight and height for age, and overall nutritional status sections were 25.1%, 47.9%, 43%, and 62.9%, respectively.

Table 5 shows the validity of S-mSTAMP using Se and Sp, PPV, NPV, and AUC as indicators, and using oSTAMP as the reference standard. The Se was excellent, but Sp was fair, when using the diagnosis section. For the nutritional intake sections, Se was good, but there was a loss of Sp. There was a good level of Se when using the weight and height for the age, but there was a loss in the Sp. The overall nutritional status shows good Se, but poor Sp.

The AUC clearly showed a good discriminative ability within the diagnosis section to determine nutritional status (Figure 1a). However, it showed a failed discriminative ability in the nutritional intake section (Figure 1b) and weight and height for age section (Figure 1c) in determining nutritional status. Also, it showed a poor discriminative ability in the overall nutritional status section to determine nutritional status (Figure 1d).

Cohen’s κ was run to determine if there was an agreement between diagnosis parts of STAMP in both the Saudi-modified and original version on whether 307 patients were at low risk of malnutrition or at high risk of malnutrition. There was moderate agreement between the two diagnosis sections in STAMP in both the Saudi-modified and original versions, κ = 0.457, *p* = 0.001. While there was no-to-slight agreement between the two nutritional intake sections, two weight and height for age sections, and two overall nutritional status sections of STAMP in both Saudi modified and original versions, the κ and *p* values came out to κ = 0.126, *p* = 0.008, κ = 0.109, *p* = 0.027, and κ = 0.089, *p* = 0.023, respectively.

### 3.6. Prevalence of Patients at High Risk of Malnutrition, and Validity of Saudi-Modified Screening Tool for the Assessment of Malnutrition in Pediatrics Using Anthropometric Reference Standards 

The Prevalence of high risk of malnourished patients based on WHO and CDC weight-for-height Z-scores, CDC BMI-for-age Z-scores, Saudi weight-for-height Z-scores, and Saudi BMI-for-age Z-scores were 31%, 28.3%, 23.2%, and 25.7%, respectively.

Table 6 shows a good Se for the overall nutritional status of S-mSTAMP when compared to the nutritional status of WHO and CDC weight-for-height Z-scores, CDC BMI-for-age Z-scores, Saudi weight-for-height Z-scores, and Saudi BMI-for-age Z-scores, increased in NPV, but reduced with a loss of specificity in PPV.

### 3.7. Prevalence of Patients at High Risk of Malnutrition and Validity of the Original Screening Tool for the Assessment of Malnutrition in Pediatrics Using Anthropometric Measurements and Dietary Intake as Reference Standard 

The prevalence of patients at high risk of malnutrition based on WHO and CDC weight-for-height Z-score, CDC BMI-for-age Z-score, Saudi weight-for-height Z-score, and Saudi BMI-for-age Z-score, were 56.7%, 28.3%, 23.2%, and 25.7%, respectively.

Table 7 shows the Se, Sp, PPV, and NPV of oSTAMP using anthropometric measurements as reference standards.

## 4. Discussion

### 4.1. Main Findings

In the current study, a validation of the Saudi-modified tool was conducted by incorporating modifications to enhance its applicability among hospitalized Saudi children. We specifically added the primary medical condition that contributes the most to mortality in Saudi children related to malnutrition, relying on information from reputable sources such as the World Health Organization (WHO) and the Center for Disease Control and Prevention (CDC), as reported by the Saudi Ministry of Health [31,32,33,34]. Additionally, we utilized Saudi growth charts (SGCs) as part of our screening processes [31,32,33,34].

The study’s main interesting finding was the prevalence of patients at high risk of malnutrition as determined by the original Screening Tool for the Assessment of Malnutrition in Pediatrics (oSTAMP) and Saudi-modified Screening Tool for the Assessment of Malnutrition in Pediatrics (S-mSTAMP) tools, which yielded rates of 59% and 91.6%, respectively. It is possible that the differences in prevalence between the two tools can be attributed to the modifications applied, which were tailored to better suit the study population; additionally, screening was applied for the patient in oSTAMP by nurses and in S-mSTAMP by dietitians, and differences in nutritional intake sections exist between the tools. The approach used to classify the nutritional status of STAMPs was similar to that employed by other researchers, wherein the moderate and high-risk (HR) categories were combined to form a group named “at high risk of malnutrition”. This group exhibited a notably high prevalence rate of 79% [35]. Another study conducted on the outpatient Egyptian population, utilizing the STAMP tool, revealed a prevalence of 24.2% of patients at high risk of malnutrition. It is possible that these results can be attributed to the STAMP tool being more suitable for inpatient populations, as mentioned previously [21].

Our study utilized weight-for-height and body mass index (BMI)-for-age Z-scores from SGCs, World Health Organization growth charts (WHOGCs), and Centers for Disease Control and Prevention growth charts (CDCGCs), aligning with previous studies that reported malnutrition prevalence rates of 19.5% and 23.2% when applying WHOGCs, weight-for-height, and BMI-for-age Z-scores, respectively [35].

However, the observed difference between the nutritional status identified using the oSTAMP and S-mSTAMP tools, and anthropometric measurements such as weight-for-height and BMI-for-age Z-scores, were not found to be statistically significant in our study. These results demonstrate comparability between the two screening tools for identifying patients at high risk of malnutrition. These results closely align with those reported by Shu Hwa Ong et al., where the STAMP tool demonstrated a prevalence rate of 35.4% for patients at high risk of malnutrition using an objective assessment that incorporated anthropometric measurements such as weight-for-height Z-score, weight-for-age Z-score, BMI-for-age Z-score, and MUAC. The assessment followed the ASPEN/AND definition of malnutrition and considered various variables [35].

However, the findings of our study do not support the previous research that demonstrated a significant association between STAMP and the WHO weight-for-height Z-score. The previous study reported a higher prevalence of malnutrition, about 73.5%, when utilizing the WHO weight-for-height Z-score as a reference standard in conjunction with STAMP for children under 2 years old [21]. This disparity may be attributed to the challenges associated with using STAMP for children within this age group, as the accuracy of nutritional intake assessments using STAMP could be compromised. Consequently, there is a potential for overestimation of the high risk of malnutrition rates when employing STAMP in this particular age bracket [21].

The results of this study align with a previous observational study, where the mean ± SD of mid-upper arm circumference (MUAC) was 159 ± 1.6, similar to the mean of 153.37 ± 40.69 in our study [35]. However, there was a notable difference in the standard deviations (SDs) between the two studies. This dissimilarity may be attributed to the fact that our study adopted a different reference for determining nutritional status based on MUAC, which included all patients up to 14 years old, in contrast to the previous study that relied on the WHO standard for patients up to 59 months old [35].

In our study, we assessed the validity and accuracy of S-mSTAMP by comparing it to the oSTAMP as a reference standard. It is encouraging to compare these results with those found by M. Reed (2020), who reported 89% sensitivity and 97% specificity for electronic health records using the original STAMP as a reference standard [36]. Additionally, they found a PPV of 60%, an NPV of 94%, and an overall accuracy of 85%. This level of sensitivity and PPV is comparable to what we observed in our study. However, Reed’s study demonstrated significantly higher specificity, accuracy, and NPV. The differences in performance may be attributed to the use of electronic health records, which can help reduce human error, as well as the implementation of staff training to ensure accurate use of both STAMP tools. Another study evaluated the efficacy of STAMP in two ways: based on the WHOGC, and based on the Hellenic growth charts (HGC) using dietetic assessment as a reference standard, which showed that the agreement of WHOGC STAMP was 0.28, while the HGC STAMP was 0.26; and Se, Sp, PPV, and NPV of WHOGC STAMP vs. HGC STAMP were 84.4% vs. 78.3 [23].

Additionally, in our study, the validities of S-mSTAMP and the oSTAMP in determining nutritional status using anthropometric measurements as reference standards are compared. It is interesting to compare the results of the current study with those of other studies that have assessed the validity of STAMP in determining nutritional status. Ong et al. found that STAMP had a fair Se of 76.32% and a poor Sp of 18.18%, with a poor NPV of 47.06%. The PPV of STAMP indicates that 45% of the children classified as at high risk of malnutrition were truly malnourished [35]. Similarly, Tuokkola et al. (2019) reported excellent Se and NPV at 100%, but a loss of Sp at 69% and a poor PPV of 17% when using anthropometric measurements as a reference standard [37]. On the other hand, Sayed et al. (2023) used weight-for-height as the gold standard, finding that STAMP had fair Se at 73.5% and good specificity at 81.4% for predicting wasting [21]. Another European study by Chourdakis et al. (2016) used height/length, weight, and BMI as reference standards and reported fair Se (79.04%), excellent NPV (90.10%), poor Sp (42.68%), and poor PPV (23.57%) [14]. Overall, these findings indicate some variability in the performance of STAMP in different studies, particularly in terms of Se, Sp, NPV, and PPV. This variability may be related to each study’s specific reference standards and different methodologies. This research project presents a significant opportunity to advance the understanding of pediatric undernutrition and enhance the effectiveness of malnutrition screening tools for hospitalized children, particularly in the context of Saudi Arabia.

### 4.2. Strengths and Limitations

This study is unique in that it not only validated the original screening tool but also a modified pediatric screening tool to better align with the characteristics of our population. By shedding light on the strengths and weaknesses of the pediatric malnutrition screening process as a clinical practice, this study aims to contribute valuable insights to the field. Since this project was conducted during the transitional phase from paper to electronic health records, the accuracy of the tool application may have been compromised, leading to potential discrepancies among the results. Furthermore, differences in anthropometric measurement assessment between the WHO/CDC GCs and SGCs, where electronic measurements were used, as opposed to human measurements, may have impacted the accuracy of the SGCs due to potential human error. It is also important to note that this study was an observational cross-sectional study conducted at a single center in the emergency ward during the winter season, limiting the generalizability of the results.

### 4.3. Implications for Practice and Future Direction

A multicenter, multi-season prospective cross-sectional study is necessary to assess the screening process upon admission and during hospitalization for children in a nationwide context. Additionally, a longitudinal prospective cohort study is recommended to evaluate the validity and agreement between different tools, such as oSTAMP and mSTAMP, on a large sample size that encompasses diverse populations at a national level. The assigned healthcare providers implementing the nutrition screening process need nutritional education, and/or should be dietitians or healthcare providers with a nutritional background, while considering the appropriate staff load to keep good performance and improve the quality of patient care. Standardizing the reference used for screening and assessments at the nationwide practice level can enhance patient outcomes by facilitating early detection of malnutrition, improving the accuracy of clinical practice, reducing the financial burden, decreasing hospital stays, and increasing bed availability.

## 5. Conclusions

This study revealed that approximately 62.9% and 91.6% of hospitalized children were identified as having a high risk of malnutrition when using oSTAMP and S-mSTAMP, respectively. Notably, S-mSTAMP demonstrated excellent sensitivity and varied in its NPV for screening malnutrition risk among hospitalized children, relying on the oSTAMP and anthropometric measurements as reference standards. Furthermore, it is crucial for hospitals and healthcare facilities to incorporate screening tools suitable for their populations into their protocols and guidelines to ensure the comprehensive screening, assessment, and management of pediatric malnutrition in hospital settings.

## Figures and Tables

**Figure 1 diagnostics-14-02256-f001:**
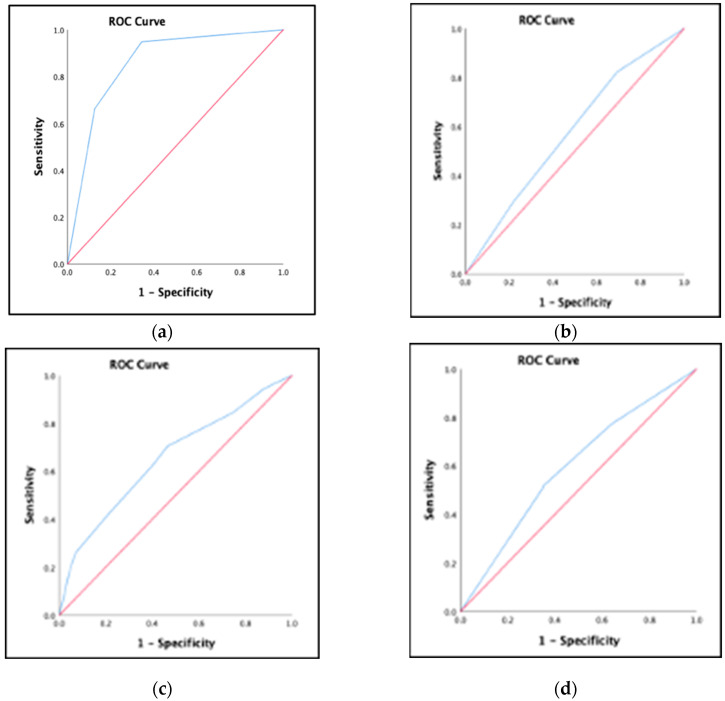
Receiver operating characteristic (ROC) curve plot of the true positive rate (sensitivity) against the false positive rate (1-specificity) (**a**) for the diagnosis section of the modified Screening Tool for the Assessment of Malnutrition in Pediatrics (mSTAMP) compared with the diagnosis section of the original Screening Tool for the Assessment of Malnutrition in Pediatrics (oSTAMP); (**b**) for nutritional intake of S-mSTAMP compared with the nutritional intake of oSTAMP; (**c**) for weight and height for age of S-mSTAMP compared with the weight and height for age of oSTAMP; (**d**) for the overall nutritional status of S-mSTAMP compared with the overall nutritional status of oSTAMP.

**Table 1 diagnostics-14-02256-t001:** Characteristics of the patients.

Patient’s Characteristics	
Wight (kg) ** All (*n* = 307)	Median & (IQR)
	15.2 (13.5)
Length/Height (cm) * All (*n* = 307)	Mean ± SD
	104.2 ± 26.73
MUAC (mm) * All (*n* = 307)	Mean ± SD
	153.37 ± 40.69
BMI (kg/m^2^) * All (*n* = 307)	Mean ± SD
	17.12 ± 4.69
Admission Diagnosis Documented *** All (*n* = 307)	Number and Percentage
Bowel failure, intractable diarrhea	11 (3.6%)
Burns and major trauma	4 (1.3%)
Liver disease	3 (1%)
Major surgery	6 (2%)
Food allergies/intolerance	4 (1.3%)
Oncology on active treatments	1 (0.3%)
Renal disease/failure	8 (2.6%)
Inborn errors of metabolism	14 (4.6%)
Coeliac disease	1 (0.3%)
Gastro-esophageal reflux	12 (3.9%)
Diabetes	9 (2.9%)
Respiratory syncytial virus	1 (0.3%)
Minor surgery	1 (0.3%)
Asthma	104 (33.9%)
Neuromuscular conditions	18 (5.9%)
Under investigation	177 (57.7%)
Pneumonia	38 (12.4%)
Acute diarrhea	12 (3.9%)
Respiratory Infection	16 (5.2%)
Neonatal disorder	7 (2.3%)

MUAC = Mid-upper arm circumference, BMI = body mass index, * data were presented as mean ± standard deviation (SD), ** data were presented as median and IQR, *** data were presented as number and percentage *n* (%).

**Table 2 diagnostics-14-02256-t002:** Nutrition Status of the Patients Based on Anthropometric Measurements.

Variables	Sample Size
	Number and Percentage
**Weight-for-height Z-score (WHO/CDC) ***	***N* = 155 (age < 59 months)**
Malnutrition	48 (31%)
Absence of malnutrition	107 (69%)
**Weight-for-height Z-score (Saudi) ***	***N* = 155 (age < 59 months)**
Malnutrition	36 (23.2%)
Absence of malnutrition	119 (76.8%)
**BMI-for-age Z-score (CDC) ***	***N* = 152 (age > 59 months)**
Malnutrition	43 (28.3%)
Absence of malnutrition	109 (71.7%)
**BMI-for-age Z-score (Saudi) ***	***N* = 152 (age > 59 months)**
Malnutrition	39 (25.7%)
Absence of malnutrition	113 (74.3%)

WHO = World Health Organization, CDC = Central of Disease Control and Prevention, BMI = body mass index, * data were presented as number and percentage *n* (%).

**Table 3 diagnostics-14-02256-t003:** Characteristics of each part of both Screening Tools for the Assessment of Malnutrition in Pediatrics.

STEPs	Original STAMP	Saudi-Modified STAMP
	Number and Percentage	
**STEP 1—DIAGNOSIS *** Does the child have a diagnosis that has any nutritional implications?		
Definite nutritional implications	51 (16.6%)	80 (26.1%)
Possible nutritional implications	27 (8.8%)	72 (23.5%)
No nutritional implications	229 (74.6%)	155 (50.5%)
**STEP 2—NUTRITIONAL INTAKE *** What is the child’s nutritional intake?		
No change in eating patterns and good nutritional intake	160 (52.1%)	75 (24.4%)
Recently decreased or poor nutritional intake	124 (40.4%)	154 (50.2%)
No nutritional intake	23 (7.5%)	78 (25.4%)
**STEP 3—WEIGHT AND HEIGHT *** Using the centile quick-reference tables to determine the child’s measurements		
0 to 1 centile spaces/columns apart	175 (57%)	94 (30.6%)
>2 centile spaces/=2 columns apart	93 (30.3%)	85 (27.7%)
>3 centile spaces/≥3 columns apart (or weight < 2nd centile)	39 (12.7%)	128 (41.7%)
**STEP 4—OVERALL RISK OF MALNUTRITION *** Add the scores from steps 1–3 together to calculate the child’s overall risk of malnutrition.		
High risk	72 (23.5%)	189 (61.5%)
Medium risk	121 (39.4%)	92 (30%)
Low risk	114 (37.1%)	26 (8.5%)
Nutritional status *		
At high risk	193 (62.9%)	281 (91.5%)
At low risk	114 (37.1%)	26 (8.5%)

STAMP = Screening Tool for the Assessment of Malnutrition in Pediatrics, * data were presented as number and percentage (%).

**Table 4 diagnostics-14-02256-t004:** Anthropometric characteristics of hospitalized children according to original and Saudi-modified Screening Tools for the Assessment of Malnutrition in Pediatrics.

Anthropometric Measurements	Nutritional Status	All	At Low Risk	At High Risk	*p*-Value
		Number and Percentage			
**Overall Original STAMP Score (*n* = 307)**					
Saudi weight-for-height Z-scores (*n* = 155) *	Malnutrition	36 (23.2%)	12 (18.7%)	24 (26.4%)	0.268 ^a^
	Absence of malnutrition	119 (76.8%)	52 (81.3%)	67 (73.6%)	
Saudi BMI-for-age Z-scores (*n* = 152) *	Malnutrition	39 (25.7%)	10 (20%)	29 (28.4%)	0.263 ^a^
	Absence of malnutrition	113 (74.3%)	40 (80%)	73 (71.6%)	
WHO and CDC weight-for-height Z-scores (*n* = 155) *	Malnutrition	48 (31%)	16 (24.6%)	32 (35.6%)	0.146 ^a^
	Absence of malnutrition	107 (69%)	49 (75.4%)	58 (64.4%)	
CDC BMI-for-age Z-scores (*n* = 152) *	Malnutrition	43 (28.3%)	13 (26.5%)	30 (29.1%)	0.740 ^a^
	Absence of malnutrition	109 (71.7%)	36 (73.5%)	73 (70.9%)	
**Overall Saudi-Modified STAMP Score (*n* = 307)**					
Saudi weight-for-height Z-scores (*n* = 155) *	Malnutrition	36 (23.2%)	2 (18.2%)	34 (23.6%)	1 ^a^
	Absence of malnutrition	119 (76.8%)	9 (81.8%)	110 (76.4%)	
Saudi BMI-for-age Z-scores (*n* = 152) *	Malnutrition	39 (25.7%)	2 (13.3%)	37 (27%)	0356 ^a^
	Absence of malnutrition	113 (74.3%)	13 (86.7%)	100 (73%)	
WHO and CDC weight-for-height Z-scores (*n* = 155) *	Malnutrition	48 (31%)	1 (9.1%)	47 (32.6%)	0.174 ^a^
	Absence of malnutrition	107 (69%)	10 (90.9%)	97 (67.4%)	
CDC BMI-for-age Z-scores (*n* = 152) *	Malnutrition	43 (28.3%)	2 (13.3%)	41 (29.9%)	0.235 ^a^
	Absence of malnutrition	109 (71.7%)	13 (86.7%)	96 (70.1%)	

STAMP = Screening Tool for the Assessment of Malnutrition in Pediatrics, WHO = World Health Organization, CDC = Centers for Disease Control and Prevention, MUAC = mid-upper arm circumference, BMI = body mass index, * data were presented as number and percentage *n* (%), ^a^—Chi-square test. *p*-value ≤ 0.05 is considered statistically significant.

**Table 5 diagnostics-14-02256-t005:** Prevalence of Patients at High Risk of Malnutrition, Validity, and Agreements of Saudi Modified Screening Tool for the Assessment of Malnutrition in Pediatrics using Original Screening Tool for the Assessment of Malnutrition in Pediatrics.

Statistical Parameters of Concurrent Validity	Diagnosis	Nutritional Intake	Anthropometrics	Overall Nutritional Status
Sensitivity	94.8%	82.3%	75.8%	94.3%
Specificity	65.7%	30.6%	36%	13.2%
Positive Predictive Value	48%	52.2%	47.2%	64.8%
Negative Predictive Value	97.4%	65.3%	66.3%	57.7%
AUC (CI 95%)	0.856 (0.810–0.902), *p* = 0.001	0.576 (0.512–0.640), *p* = 0.021	0.595 (0.531–0.659), *p* = 0.04	0.654 (0.592–0.715), *p* = 0.001

AUC = Area Under Curve, CI = confidence interval. *p*-value ≤ 0.05 is considered statistically significant.

**Table 6 diagnostics-14-02256-t006:** Prevalence of Patients at High Risk of Malnutrition, and Validity of Saudi Modified Screening Tool for the Assessment of Malnutrition in Pediatrics using Anthropometric Measurements as Reference Standard.

Statistical Parameters of Concurrent Validity	WHO and CDC Weight-for-Height Z-Score	CDC BMI-for-Age Z-Score	Saudi Weight-for-Height Z-Score	Saudi BMI-for-Age Z-Score
Sensitivity	97.9%	95.3%	94.4%	94.9%
Specificity	9.3%	11.9%	7.6%	11.5%
Positive Predictive Value	32.6%	29.9%	23.6%	27%
Negative Predictive Value	90.9%	86.7%	81.8%	86.7%

WHO = World Health Organization, CDC = Central of Disease Control and Prevention, BMI = body mass index.

**Table 7 diagnostics-14-02256-t007:** Prevalence of Patients at High Risk of Malnutrition and Validity of The Original Screening Tool for the Assessment of Malnutrition in Pediatrics using Anthropometric Measurements as Reference Standard.

Statistical Parameters of Concurrent Validity	WHO and CDC Weight-for-Height Z-Score	CDC BMI-for-Age Z-Score	Saudi Weight-for-Height Z-Score	Saudi BMI-for-Age Z-Score
Sensitivity	54.2%	69.8%	66.7%	74.4%
Specificity	33.3%	33%	43.7	35.4%
Positive Predictive Value	64.4%	29%	26.4%	28.4%
Negative Predictive Value	24.6%	73.5%	81.3%	80%

STAMP = Screening Tool for the Assessment of Malnutrition in Pediatrics, WHO = World Health Organization, CDC = Central of Disease Control and Prevention, BMI = body mass index.

## Data Availability

The data supporting this study’s findings are available from the corresponding author upon request.

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
