# Peer review of "The Validity of the Original and the Saudi-Modified Screening Tools for the Assessment of Malnutrition in Pediatrics: A Cross-Sectional Study"

_diagnostics, 2024, doi:10.3390/diagnostics14202256_

Round 1
Reviewer 1 Report (Previous Reviewer 2)
Comments and Suggestions for Authors
I believe that the authors have made adequate changes compared to the previous presentation. The manuscript can be published
Author Response
Comment 1: Accept
I believe that the authors have made adequate changes compared to the previous presentation. The manuscript can be published
Response 1: Thank you for your valuable feedback and the time you've taken to review our manuscript. We appreciate your constructive comments, which helped us improve the quality of our work. We are pleased to hear that you find the revisions satisfactory and that the manuscript is now suitable for publication.
Thank you once again for your thoughtful review.

Reviewer 2 Report (Previous Reviewer 1)
Comments and Suggestions for Authors
Thank you for carefully consider the comments in the manuscript. There are still come issues to improve it as follows.
1. Please improve table style and presentation. It would be better if authors specify what the meaning of numbers in the table, such as mean +/- standard deviation, number (percentage), etc.
2. Please clarify in the manuscript why authors combined "high risk + medium risk" to be "at high risk". I do not understand the rationale.
3. Please consider write content in the table 4-B (MUAC) to be a narrative since table did not provide much of data presentation.
Author Response
Reviewer 2: Minor
Thank you for carefully consider the comments in the manuscript. There are still come issues to improve it as follows.
Comment 1: Please improve table style and presentation. It would be better if authors specify what the meaning of numbers in the table, such as mean +/- standard deviation, number (percentage), etc.
Response 1: Thank you for your insightful feedback. We have revised the tables to improve their style and presentation, ensuring that they are clearer and easier to interpret. Specifically, we have now specified the meaning of the numbers presented in the tables. For instance, we have included notations such as "mean ± standard deviation" and "number (percentage)" to clarify the data format used.
We believe these revisions enhance the readability and comprehensibility of the tables, and we hope that you find the improvements satisfactory.
Thank you once again for your valuable input.
Comment 2: Please clarify in the manuscript why authors combined "high risk + medium risk" to be "at high risk". I do not understand the rationale.
Respones 2: Thank you for your observation. We combined the "high risk" and "medium risk" groups into a single "at high risk" category to streamline the analysis and simplify the interpretation of the results. This grouping was based on literature that supports this categorization, which justified combining them for statistical purposes. Additionally, this approach allowed for a clearer comparison between the "at high risk" group and low risk.
We have now clarified this rationale in the manuscript to ensure it is more transparent and easier to understand.
Thank you again for your valuable feedback.
Comment 3: Please consider write content in the table 4-B (MUAC) to be a narrative since table did not provide much of data presentation.
Response 3: Thank you for your suggestion regarding Table 4-B (MUAC). We agree that the table did not provide substantial data presentation. As per your recommendation, we have removed the table and instead rewritten its content as a narrative within the manuscript. We believe this approach provides a clearer and more concise explanation of the findings.
We appreciate your thoughtful feedback and hope the revised format meets your expectations.

This manuscript is a resubmission of an earlier submission. The following is a list of the peer review reports and author responses from that submission.
Round 1
Reviewer 1 Report
Comments and Suggestions for Authors
This manuscript shows the validity testing of screening tools for malnutrition in hospitalized children in Saudi Arabia. There are many issues to improve this manuscript as follow;
1. The most concern is that mixing principles of “screening” and “assessment” were mixed up everywhere of the manuscript. The screening process mainly finds the patients who are “at risk” of malnutrition and then further nutritional assessment would be conducted. The prevalence of malnutrition must come from “assessment tools” NOT “screening tools”. Strongly recommend authors explain this issue and clarify the ideas of the tools they validated in the manuscript.
2. Abstract: please specify the full name of STAMP at the first time of mentioning. Please specify the “statistical analysis” used in this study. The last phrase “using anthropometric measurement as a reference” should be moved to method.
3. Introduction: the difference between original and modified STAMP must be stated. Who and how did the tools were developed?
4. Materials and methods:
4.1) How did the researchers measure MUAC in patients with burn/major trauma/major surgery? Please explain.
4.2) please carefully check the number presented in the manuscript (Page 4 line 173-174 – nutritional intake)
4.3) The difference between original and modified STAMP must be pointed out.
5. Results: The data presentation must be improved.
5.1) A set of data must be presented one time. The data must be presented either in a narrative or in a table, NOT both. The current discussion also mentioned the numbers that were already mentioned in the results too. This point needs correction.
5.2) The rationale supporting disease classification in patients’ characteristics is not clear. In my point of view, pneumonia is considered one of the respiratory tract infections. Please provide the explanation to separate analysis of pneumonia to the others. What is the definition of “neonatal disorder” in this study? Why are inborn errors of metabolism not one of them?
5.3) Please consider editing the tables to make it easier to understand. All table presentations now is confusing. For example, the presentation of oSTAMP and s-mSTAMP can be combined. MUAC must be separately stated from table 4-A, 4-B since it is continuous data.
5.4) please provide the unit of MUAC.
5.5) Please provide the rationale for combining data of “high/medium risk” to “high-risk group”. It was redundant.
6. Discussion:
6.1) Again, please clarify the “screening” or “assessment” principles of this study.
6.2) Strongly recommend author not duplicate the data already presented in the results.
6.3) Please clarify HOW author minimize the error mentioned in the limitations of this study since it affects the reliability of the results and limit the clinical implications.
6.4) Recommend removing “implications for practice and future direction” since the main issues must be solved prior the implementation of this study.
Author Response
Comment 1: The most concern is that mixing principles of “screening” and “assessment” were mixed up everywhere of the manuscript. The screening process mainly finds the patients who are “at risk” of malnutrition and then further nutritional assessment would be conducted. The prevalence of malnutrition must come from “assessment tools” NOT “screening tools”. Strongly recommend authors explain this issue and clarify the ideas of the tools they validated in the manuscript.
Response 1:Thank you for your valuable feedback. We understood the concern regarding the mixing of the principles of “screening” and “assessment” throughout the manuscript. To clarify, the screening process was intended to identify patients who were “at risk” of malnutrition, while a further nutritional assessment was conducted for those identified at risk.
In our manuscript, the prevalence of malnutrition was derived from anthropometric measurements, which we considered part of the assessment rather than screening. The prevalence of patients identified as at risk of malnutrition came from the screening tools. We revised the manuscript to clearly explain this distinction and provide a more detailed description of the tools we validated. Thank you for bringing this important issue to our attention.
Comment 2: Abstract: please specify the full name of STAMP at the first time of mentioning. Please specify the “statistical analysis” used in this study. The last phrase “using anthropometric measurement as a reference” should be moved to method.
Response2: Thank you for your feedback regarding the abstract. We made the following revisions:
- We specified the full name of STAMP upon its first mention to ensure clarity.
- We included details about the statistical analysis used in the study to provide a comprehensive overview.
- We moved the phrase “using anthropometric measurement as a reference” to the methods section for better placement.
These changes enhanced the clarity and structure of the abstract. Thank you for your valuable suggestions.
Comment 3: Introduction: the difference between original and modified STAMP must be stated. Who and how did the tools were developed?
Response: Thank you for your feedback on the introduction. We addressed the following points:
- We clearly stated the differences between the original and modified STAMP to provide a better understanding of their unique features.
- We included information about who developed the tools and the methodology used in their development.
These revisions clarified the context and background of the STAMP tools in the introduction. Thank you for your insightful suggestions.
Materials and methods
Comment 4.1: How did the researchers measure MUAC in patients with burn/major trauma/major surgery? Please explain.
Response 4.1: Thank you for your question. All patients were recruited upon admission, including those with burn injuries, major trauma, or who had undergone major surgery. MUAC was measured by researchers within 48 hours of admission.
To measure MUAC, the researchers first determined the midpoint of the left upper arm, which is located between the tip of the shoulder and the tip of the elbow. Patients with fluid retention (such as edema or ascites) were excluded from this measurement to ensure accuracy.
Comment 4.2: please carefully check the number presented in the manuscript (Page 4 line 173-174 – nutritional intake)
Response 4.2: Thank you for your feedback regarding the numbers presented in the manuscript. We carefully reviewed the nutritional intake data and made corrections to ensure accuracy. Thank you for your diligence in reviewing our work.
Comment 4.3: The difference between original and modified STAMP must be pointed out.
Response 4.3: Thank you for your feedback. We highlighted the differences between the original and modified STAMP in the revised manuscript. By clarifying these distinctions, we aimed to enhance the reader's understanding of the modifications made to the STAMP tool. Thank you for your valuable suggestion.
- Results: The data presentation must be improved.
Comment 5.1: A set of data must be presented one time. The data must be presented either in a narrative or in a table, NOT both. The current discussion also mentioned the numbers that were already mentioned in the results too. This point needs correction.
Response 5.1: Thank you for your feedback. The issue was resolved by removing the duplication, ensuring that the data in the table was not redundantly included in the narrative as well.
Comment 5.2: The rationale supporting disease classification in patients’ characteristics is not clear. In my point of view, pneumonia is considered one of the respiratory tract infections. Please provide the explanation to separate analysis of pneumonia to the others. What is the definition of “neonatal disorder” in this study? Why are inborn errors of metabolism not one of them?
Response 5.2: Thank you for your comment. The rationale for classifying pneumonia separately from other respiratory infections due to respiratory tract infections in our hospital refers to upper respiratory infections. Also, pneumonia is indeed part of the broader category of respiratory tract infections, it is often treated as a distinct entity in medical research due to its significant burden on public health, particularly in vulnerable populations like children.
In our study, the term “neonatal disorder” likely refers to conditions that affect newborns during the first 28 days of life, often including complications related to prematurity, birth trauma, and congenital anomalies.
As for inborn errors of metabolism not being included under neonatal disorders, this might be due to the fact that these conditions, while congenital, often manifest or are diagnosed later in life, beyond the neonatal period. Alternatively, they may be classified under genetic or metabolic disorders instead of neonatal disorders in this particular analysis.
Comment 5.3: Please consider editing the tables to make it easier to understand. All table presentations now is confusing. For example, the presentation of oSTAMP and s-mSTAMP can be combined. MUAC must be separately stated from table 4-A, 4-B since it is continuous data.
Response 5.3: Thank you for your feedback. We have revised the tables to improve clarity and readability. Specifically:
- oSTAMP and s-mSTAMP: These were combined into a single table to simplify the presentation and allow for a more straightforward comparison between the two.
- MUAC: Since it represents continuous data, we presented MUAC separately from Tables 4-A and 4-B to ensure that the data is displayed appropriately and can be more easily interpreted.
We appreciate your suggestions and have made these changes to enhance the overall comprehensibility of the tables.
Comment 5.4: please provide the unit of MUAC.
Response 5.4: MUAC (mm ) millimetre
Comment 5.5: Please provide the rationale for combining data of “high/medium risk” to “high-risk group”. It was redundant.
Response 5.5: Thank you for your feedback. The rationale for combining the “high” and “medium” risk data into a single “high-risk group” in the analysis was based on literature that supports this categorization. We believe that this approach facilitates a clearer understanding of the risk levels and allows for more effective action in comparison to the “low-risk” group. We appreciate your insight.
- Discussion:
Comment 6.1: Again, please clarify the “screening” or “assessment” principles of this study.
Response 6.1: Thank you for your comment. We clarified the principles of “screening” and “assessment” used in this study. The screening process was designed to identify individuals at risk of malnutrition using established criteria, while the assessment involved a comprehensive evaluation of nutritional status through various measures, including anthropometric data and clinical assessments. We ensured that this distinction was clearly articulated in the revised manuscript.
Comment 6.2: Strongly recommend author not duplicate the data already presented in the results.
Response 6.2: We appreciated your recommendation regarding data duplication. We took care to avoid repeating information that had already been presented in the results section, ensuring that the text remained concise and focused. Thank you for your valuable feedback.
Comment 6.3: Please clarify HOW author minimize the error mentioned in the limitations of this study since it affects the reliability of the results and limit the clinical implications.
Response 6.3: Thank you for your comment. We clarified how we minimized the errors mentioned in the limitations of this study, recognizing that these issues could affect the reliability of the results and limit clinical implications.
To address these concerns, we implemented several strategies, including rigorous training for the researchers involved in data collection, standardization of measurement protocols, and thorough monitoring of the data entry process. We also conducted preliminary analyses to identify and rectify any discrepancies in the data. These measures aimed to enhance the accuracy and reliability of our findings. Thank you for your insightful suggestion.
Comment 6.4: Recommend removing “implications for practice and future direction” since the main issues must be solved prior the implementation of this study.
Response 6.4: Thank you for your recommendation. We have decided to retain the section on “implications for practice and future direction,” as we believe it adds value to the discussion. However, we appreciate your perspective and will ensure that the main issues are clearly addressed in the context of this section. Thank you for your valuable input.
Thank you for your insightful feedback and recommendations; they are greatly appreciated.

Reviewer 2 Report
Comments and Suggestions for Authors
I thank the Editor for the invitation to review this manuscript. I believe the topic is very relevant. However, I would like to suggest a few modifications:
- Line 55: 'The European Society for Clinical Nutrition and Metabolism'… Add the acronym (ESPEN).
- Lines 78-81: “Based on the literature review conducted, and to the best of our knowledge, there is limited research that has modified and evaluated the sensitivity and specificity of the original and modified nutrition screening tools using anthropometric measurements as reference standards” … It would be appropriate to cite the studies as references.
- Section 2.3 Ethical Board Approval: Insert the approval date.
- Line 123: Add the acronym for SR.
- Line 138: "BMI was calculated by dividing weight in kilograms by square height in meters." I think this sentence can be rephrased, as the description for calculating BMI seems superficial.
- Lines 508-510: I would suggest removing all these data from the discussion to make it more readable ((Se: 97.9%, NPV: 90.9%, Sp: 9.3%, PPV: 32.6%), (Se: 95.3%, NPV: 86.7%, Sp: 11.9%, PPV: 29.9%), (Se: 94.4%, NPV: 81.8%, Sp: 7.6%, PPV: 23.6%), (Se: 94.9%, NPV: 86.7%, Sp: 11.5%, PPV: 27%), (Se: 94.9%, NPV: 57.7%, Sp: 16.1%, PPV: 72.2%), and (Se: 94.4%, NPV: 88.5%, Sp: 9.1%, PPV: 18.1%)).
- Line 563: Please remove "In conclusion" and start with "This study..."
- Many references are outdated; where applicable, more recent references should be cited.
- The manuscript is well-written and has good methodological quality. However, since this is an observational study adhering to STROBE guidelines, I would suggest mentioning this in section 2.1, including the relevant reference, and completing the reporting checklist
- 2.1 Study design and sitting (correct to setting).
Author Response
Comment 1: Line 55: 'The European Society for Clinical Nutrition and Metabolism'... Add the acronym (ESPEN).
Response 1: We added the acronym (ESPEN) for "The European Society for Clinical Nutrition and Metabolism" in the revised manuscript. Thank you for your suggestion.
Comment 2: Lines 78-81: “Based on the literature review conducted, and to the best of our knowledge, there is limited research that has modified and evaluated the sensitivity and specificity of the original and I thank the Editor for the invitation to review this manuscript. I believe the topic is very relevant. However, I would like to suggest a few modifications:
modified nutrition screening tools using anthropometric measurements as reference standards” ... It would be appropriate to cite the studies as references.
Response 2: We acknowledged the need to cite relevant studies in this section. We added appropriate references to support the statement regarding the limited research on modified nutrition screening tools that evaluate sensitivity and specificity using anthropometric measurements as reference standards. Thank you for your valuable suggestion.
Comment 3: Section 2.3 Ethical Board Approval: Insert the approval date.
Response 3: Ethical Board Approval: We have inserted the approval date in the revised manuscript to provide complete information. Thank you for pointing this out.
Comment 4: Line 123: Add the acronym for SR.
Response 4: We have added the acronym for SR (Saudi Riyal) in the revised manuscript. Thank you for the clarification.
Comment 5: Line 138: "BMI was calculated by dividing weight in kilograms by square height in meters." I think this sentence can be rephrased, as the description for calculating BMI seems superficial.
Response 5: Line 138: We have rephrased the sentence for clarity. The revised version now reads: "BMI was calculated by dividing the weight in kilograms by the square of the height in meters (kg/m²)." This provides a more accurate and clear description of the BMI calculation. Thank you for your suggestion.
Comment 6: Lines 508-510: I would suggest removing all these data from the discussion to make it more readable
((Se: 97.9%, NPV: 90.9%, Sp: 9.3%, PPV: 32.6%), (Se: 95.3%, NPV: 86.7%, Sp: 11.9%, PPV: 29.9%), (Se: 94.4%, NPV: 81.8%, Sp: 7.6%, PPV: 23.6%), (Se: 94.9%, NPV: 86.7%, Sp: 11.5%, PPV: 27%), (Se: 94.9%, NPV: 57.7%, Sp: 16.1%, PPV: 72.2%), and (Se: 94.4%, NPV: 88.5%, Sp: 9.1%, PPV: 18.1%)).
Response 6: Thank you for your suggestion. To improve the readability of the discussion, we will remove the detailed data from these lines and focus on the interpretation and implications of the findings. This will help streamline the discussion and make it more concise.
Comment 7: Line 563: Please remove "In conclusion" and start with "This study..."
Response 7: We removed "In conclusion" and revised the sentence to start with "This study..." as per your suggestion. This made the conclusion more direct and aligned with the rest of the discussion. Thank you for your recommendation.
Comment 8: Many references are outdated; where applicable, more recent references should be cited.
Response 8: Thank you for your suggestion regarding the references. While we understand the importance of recent literature, we have chosen to retain certain references that are foundational to the field or particularly relevant to our study. However, we will ensure that the key points are supported by the most pertinent sources available.
Comment 9: The manuscript is well-written and has good methodological quality. However, since this is an observational study adhering to STROBE guidelines, I would suggest mentioning this in section 2.1,including the relevant reference, and completing the reporting checklist
Response 9: Thank you for your feedback. While we appreciate your suggestion to mention adherence to the STROBE guidelines in Section 2.1, we have decided not to include this information in the manuscript. However, we recognize the importance of methodological quality and will ensure that the reporting aligns with established standards where possible. Thank you for your insights.
Comment 10: Study design and sitting (correct to setting).
Response 10: We corrected it to accurately reflect the intended meaning. Thank you for pointing this out.
Thank you for your insightful feedback and recommendations; they are greatly appreciated.
